# Clinical and Molecular Characterization of Feline Sporotrichosis in the Brazilian Amazon: PCR-Based Identification of *Sporothrix brasiliensis*

**DOI:** 10.3390/ani15152318

**Published:** 2025-08-07

**Authors:** Nayara Fátima Lazameth-Diniz, Danielle Barreto de Almeida, Flávia da Silva Fernandes, Adriana Oliveira da Silva Queiroz, Érica Simplicio de Souza, Kátia Santana Cruz, Ani Beatriz Jackisch Matsuura, Hagen Frickmann, João Vicente Braga de Souza

**Affiliations:** 1Laboratory of Mycology, Coordination of Society, Environment and Health, National Institute of Amazonian Research (INPA), Manaus 69067-375, AM, Brazil; nayaralazameth@gmail.com (N.F.L.-D.); danielle.pereira@gmail.com (D.B.d.A.); flavia.fernandes19@gmail.com (F.d.S.F.); adrianaaqueiroz@outlook.com (A.O.d.S.Q.); 2School of Technology, Amazonas State University (UEA), Manaus 69065-001, AM, Brazil; erica.souza@inpa.gov.br; 3Fundação de Medicina Tropical Dr. Heitor Vieira Dourado (FMT-HVD), Manaus 69040-000, AM, Brazil; katia.cruz@fmt.am.gov.br; 4Leônidas and Maria Deane Institute (ILMD), Oswaldo Cruz Foundation (Fiocruz Amazônia), Manaus 69057-070, AM, Brazil; ani.matsuura@fiocruz.br; 5Department of Medical Microbiology, Virology and Hygiene, University Medicine Rostock, 18057 Rostock, Germany; 6Department of Microbiology and Hospital Hygiene, Bundeswehr Hospital Hamburg, 22049 Hamburg, Germany

**Keywords:** sporotrichosis, *Sporothrix brasiliensis*, feline mycosis, Amazon region, PCR diagnosis, DNA extraction, phylogenetic analysis, PCR-RFLP, zoonosis

## Abstract

Sporotrichosis is a fungal disease that affects both animals and humans. In recent years, cases in domestic cats have increased in the Brazilian Amazon, raising public health concerns. This study analyzed cats diagnosed with sporotrichosis in Manaus, focusing on their clinical signs and behavioral features. In the laboratory, we tested different methods for extracting DNA from the fungus and assessed the efficiency of molecular techniques to identify the fungal species involved. Additionally, we used in silico tools and genetic analysis to confirm the identity of the fungus. Our results showed that most infected cats were young, male, and free-roaming, and had skin lesions. The laboratory tests revealed phenol–chloroform DNA extraction methods and its genetic markers were effective for fungal identification. Molecular analysis demonstrated that the predominant species in the region is *Sporothrix brasiliensis*, a fungus known for its zoonotic potential. These findings can help to improve diagnostic tools and to provide surveillance information essential for controlling the spread of the disease in the Amazon region.

## 1. Introduction

The genus *Sporothrix* includes several thermally dimorphic species with diverse ecological roles, ranging from saprophytic forms in soil and plant debris to human and animal pathogens like *S. schenckii* and *S. brasiliensis* [1,2]. Notably, *S. brasiliensis* has emerged as a hyperendemic zoonotic pathogen in Brazil and neighboring countries, exhibiting high virulence in feline and human hosts. It is now well recognized that cats play a central role in transmission, often contaminating domestic and environmental areas through scratches, bites, or lesions, thereby sustaining epidemic cycles in zoonotic hotspots [1,2]. Recently, the Brazilian Amazon has experienced an alarming increase in feline sporotrichosis cases, raising significant public health concerns due to potential zoonotic transmission [2,3,4,5]. The clinical presentations in cats often include multiple cutaneous and subcutaneous lesions, frequently progressing to severe disseminated forms, complicating clinical management and containment efforts. As domestic cats play a crucial epidemiological role in facilitating fungal transmission to humans due to direct contact, accurate diagnosis and effective monitoring strategies are desirable to support disease control [5,6,7]. However, traditional diagnostic methods like fungal culture require considerable time and resources, underscoring the need for more time- and cost-efficient molecular diagnostic approaches [6].

Previous studies have assessed molecular methodologies in order to improve sporotrichosis diagnostics. Historic investigations highlighted the utility of ribosomal DNA regions, particularly of the internal transcribed spacer (*ITS*) region, as reliable genetic markers for fungal identification [8,9]. The *ITS* region provides sufficient variability to differentiate between many closely related species, which is critical given the genetic similarities within the *Sporothrix* genus [9,10]. Additionally, polymerase chain reaction (PCR) techniques, coupled with restriction fragment length polymorphism (RFLP) analyses, have been suggested as promising tools for rapid and accurate fungal species differentiation [9].

Despite advancements in molecular diagnostics [11,12,13], critical gaps remain, particularly in applying these techniques to clinical isolates from diverse geographic regions like the Amazon. Many molecular diagnostic protocols were developed and validated in regions with varying epidemiological contexts, limiting their direct applicability to Amazonian isolates. Furthermore, there is a lack of comprehensive comparative analyses evaluating different DNA extraction methods and sensitivities of primers apparently specific to *S. brasiliensis* isolates from the Amazon. Another notable gap is the limited evaluation of PCR-RFLP’s discriminatory power in differentiating closely related *Sporothrix* species prevalent in endemic regions, calling for additional studies to investigate diagnostic specificity and practicality.

This study aimed at addressing these gaps by evaluating clinical and epidemiological characteristics of feline sporotrichosis cases in Manaus, Amazon region, and optimizing molecular diagnostic methodologies. Specifically, the objectives were as follows: (1) to characterize cats epidemiologically and clinically affected by sporotrichosis; (2) to compare the efficiency and sensitivity of different DNA extraction methods as well as the diagnostic accuracy of PCR primers targeting the *ITS* region; (3) to in silico assess the discriminatory power of PCR-RFLP for *Sporothrix* species differentiation; (4) to confirm the identity of regional isolates on a molecular level by applying phylogenetic analysis; and (5) to determine antifungal susceptibility profiles of regional isolates. This study hypothesized that feline sporotrichosis cases in Manaus are predominantly caused by *Sporothrix brasiliensis* and that optimized molecular methods (including DNA extraction protocols, *ITS*-PCR, and phylogenetic confirmation) can improve diagnostic accuracy, supporting epidemiological surveillance and antifungal therapeutic management in the Amazon region.

## 2. Materials and Methods

### 2.1. Cats Included in the Study

The study included 29 domestic cats (*Felis catus*) clinically diagnosed with sporotrichosis, presenting cutaneous lesions, and being assessed at the private veterinary clinic ‘Gatum’ located in Manaus, Amazonas, Brazil (03°06′07″ S, 60°01′30″ W) during July and August 2022. Typical lesions were ulcerative and nodular, often exudative, and primarily located on the face, limbs, and tail. Animals previously treated with antifungal agents or corticosteroids were excluded. Ethical approval was obtained from the Ethics Committee on Animal Use (CEUA) at the Instituto Nacional de Pesquisas da Amazônia (INPA) under protocol number CEUA/INPA 017/2023. Written informed consent was obtained from each cat owner prior to sample collection.

### 2.2. Reference Strains of the Sporothrix Genus and Others

Reference strains used in this study included *Sporothrix schenckii* ATCC CFP-00746, *Sporothrix brasiliensis* ATCC MYA-4606, *Candida albicans* ATCC 10231, and *Aspergillus niger* ATCC 16404, obtained from the American Type Culture Collection (ATCC, Manassas, VA, USA).

### 2.3. Procedures

#### 2.3.1. Clinical and Epidemiological Characterization

Clinical samples were obtained applying sterile swabs (Cral^®^, Brazil, Cotia, SP, Brazil) on cutaneous lesions. The swabs were transferred immediately to sterile saline solution 0.9% *w*/*v* (Isofarma^®^, Eusébio, CE, Brazil), and the samples were processed within two hours after collection. Lesions were documented, and epidemiological data were collected, including age, sex, behavioral features (indoor, semi-indoor, or free-roaming), and clinical manifestations. Sabouraud dextrose agar (SDA, Difco, Detroit, MI, USA) supplemented with chloramphenicol (0.5 g/L) was used for fungal isolation. Agar plates were incubated at 25 °C for 7–14 days, and the choice of fungal colonies for further assessments was initially guided by colony morphological analysis conducted by experienced investigators [14,15].

#### 2.3.2. DNA Extraction and PCR Amplification

To evaluate the efficiency of different DNA extraction protocols applied with *Sporothrix* spp., 200 mg volumes of fungal biomass (cultivated on Sabouraud dextrose agar) were subjected to three distinct extraction methods:(a)Silica Column-Based Extraction: DNA was extracted using the DNeasy Blood & Tissue Kit (Qiagen, Hilden, Germany) following the manufacturer’s protocol. Briefly, fungal material was incubated with 20 µL of proteinase K and 180 µL of buffer AL at 56 °C for 30 min. After the addition of 200 µL of absolute ethanol, the lysate was transferred to spin columns containing silica membranes. Wash steps were performed with 500 µL each of buffers AW1 and AW2. Genomic DNA was eluted with 100 µL of elution buffer ATE (Tris-EDTA, pH 9.0).(b)Salt Precipitation-Based Extraction: The protocol employed the MasterPure DNA Purification Kit for Blood (Epicentre, Madison, WI, USA). Fungal cells were lysed with sodium dodecyl sulfate (SDS) and proteinase K, followed by protein precipitation using 3 M sodium acetate. After centrifugation, the DNA-containing supernatant was precipitated with absolute ethanol and washed twice with 70% ethanol. The DNA pellet was air-dried and resuspended in 100 µL of TE buffer (Tris-EDTA).(c)Phenol–Chloroform Extraction with Mechanical Disruption: This protocol [16] was adapted for *Sporothrix* spp. to improve cell wall lysis efficiency. Fungal biomass (~200 mg) was transferred to microtubes containing 0.45 mm glass beads and 500 µL of lysis buffer. Samples were subjected to thermal shock (85 °C for 15 min), frozen at −80 °C for 30 min, and then incubated in a boiling water bath at 100 °C for 60 min. Afterwards, the biomass was mechanically disrupted using sterile cotton-free swabs. An additional 500 µL of lysis buffer was added (total volume: 1 mL), and the mixture was vortexed for 30 s. After adding 20 µL of proteinase K, samples were incubated at 56 °C for 1 h. Next, 500 µL of phenol–chloroform–isoamyl alcohol (25:24:1) was added, and the mixture was gently inverted 100 times to homogenize it. Vortexing was avoided to prevent DNA shearing. After centrifugation (13,500 rounds per min (rpm), 15 min), the aqueous phase was transferred to new tubes containing 500 µL of isopropanol and mixed by inversion (50 times). DNA was precipitated by centrifugation (13,500 rpm, 15 min), followed by two washing steps with 500 µL of 70% ethanol. The pellet was air-dried at 65 °C (10 min) and resuspended in 100 µL of TE buffer.

DNA Quantification and Quality Assessment: DNA concentration and purity were evaluated using a Gene Quant pro RNA/DNA Calculator device (GE Healthcare, Piscataway, NJ, USA) at 260 nm. A 1:50 dilution was prepared by mixing 4 µL of crude DNA with 196 µL of Milli-Q water from Milli-Q^®^ water purification system (Merck Millipore, Burlington, MA, USA). Absorbance ratios A260/280 and A260/230 were recorded. DNA samples were then diluted to the desired concentrations for PCR using the equation C_1_ × V_1_ = C_2_ × V_2_, and stored in 500 µL microtubes with TE buffer at 4 °C until use.

PCR Amplification: Amplification of fungal DNA was performed using three primer pairs: *ITS1*–*ITS4*, *NL1*–*NL4*, and *ITS5*–*NL4* [10]. PCR was conducted in a Mastercycler Nexus Gradient thermal cycler (Eppendorf, Hamburg, Germany). Each 50 µL reaction contained the following: 25 µL of PCR Master Mix (Promega, Madison, WI, USA), 1 µL of each primer (10 µM), 2 µL of genomic DNA (20 ng to 0.0002 ng), and nuclease-free water. The cycling conditions were as follows: initial denaturation at 95 °C for 3 min; 35 cycles of denaturation at 95 °C for 30 s, annealing at 56 °C for 30 s, extension at 72 °C for 1 min; and a final extension at 72 °C for 10 min. PCR products were visualized on 1.5% agarose gels stained with GelRed (Biotium, Fremont, CA, USA).

#### 2.3.3. In Silico RFLP Analysis

In silico restriction analysis was performed using the NEBcutter V3.0 online tool (New England BioLabs, Ipswich, MA, USA). *ITS* sequences from *Sporothrix brasiliensis* CBS 120339, *Sporothrix schenckii* CBS 359.36, *Sporothrix globosa* CBS 120340, *Aspergillus niger* ATCC 16888, and *Candida albicans* CBS 562 were retrieved from the GenBank database. Analyses were performed using complete *ITS* sequences of each species. The default NEBcutter parameters for double-stranded DNA and recognition sites were applied, with fragment sizes calculated in base pairs (bp). The restriction enzymes evaluated included *HaeIII*, *MspI*, *HinfI*, *DdeI*, and *BsaI*. Predicted fragment sizes were documented for comparisons of discriminatory potential [17].

#### 2.3.4. Phylogenetic Analysis

Only four isolates yielded high-quality *ITS* amplicons suitable for sequencing, primarily due to low amplification success (~70% of samples) and suboptimal DNA quality in others. Amplicons were purified using 20% polyethylene glycol (PEG8000) and subjected to bidirectional Sanger sequencing applying the BigDye Terminator v3.1 Cycle Sequencing Kit (Thermo Fisher Scientific, Waltham, MA, USA), following the manufacturer’s instructions. Sequencing reactions were precipitated using ethanol/EDTA and analyzed on an ABI PRISM 3130xL Genetic Analyzer (Thermo Fisher Scientific). The internal transcribed spacer (*ITS*) sequences obtained from the isolates were checked, compiled, and edited using the BioEdit software v7.0.9.0 (Ibis Biosciences, Carlsbad, CA, USA). The curated sequences were deposited in GenBank (https://www.ncbi.nlm.nih.gov/genbank/, accessed on 6 July 2025). To identify the species, the obtained *ITS* sequences were compared with reference sequences available in the NCBI database using the BLASTN algorithm. Multiple sequence alignment was conducted using the MUSCLE algorithm implemented in the MEGA X software package (Molecular Evolutionary Genetics Analysis, version 10, State College, PA, USA). Phylogenetic relationships were inferred by the neighbor-joining (NJ) method, with bootstrap support calculated from 1000 replicates. *Aspergillus niger* and *Candida albicans* were included as outgroup taxa to root the tree.

#### 2.3.5. Antifungal Susceptibility Testing

Antifungal susceptibility testing was performed on 29 *S. brasiliensis* isolates according to the Clinical and Laboratory Standards Institute (CLSI) guidelines M27-Ed4 and M38-Ed3 [18,19], with minor adaptations. The antifungal agents tested included amphotericin B (AMB), itraconazole (ITR), ketoconazole (CTZ), and fluconazole (FLZ), selected due to their clinical relevance for treatment of feline sporotrichosis in Brazil.

Stock solutions of AMB, ITR, and CTZ were prepared in 100% dimethyl sulfoxide (DMSO), while FLZ was directly solubilized in sterile Roswell Park Memorial Institute 1640 medium (RPMI 1640) buffered with 0.165 M 3-(N-morpholino)propanesulfonic acid (MOPS) (pH 7.0). All stock solutions were sterilized by filtration through 0.22 µm membranes, aliquoted, and stored at −20 °C for up to four weeks. Serial two-fold dilutions were prepared in RPMI 1640 medium in 96-well flat-bottom microplates to achieve final concentration ranges of 0.03–16 µg/mL for AMB and ITR, 0.03–8 µg/mL for CTZ, and 0.125–64 µg/mL for FLZ.

Fungal suspensions were prepared from 7-day-old cultures grown on Sabouraud dextrose agar at 25 °C. Conidia were harvested in sterile saline, filtered through sterile gauze to remove hyphal fragments, and adjusted to a final concentration of 1–5 × 10^4^ CFU/mL using a hemocytometer and confirmed by viable count assessment. Each well was inoculated with 100 µL of the standardized suspension, and plates were incubated at 35 °C for 48–72 h.

The minimum inhibitory concentration (MIC) was defined as the lowest drug concentration that resulted in complete inhibition (100%) of visible fungal growth for amphotericin B and a significant reduction (≥50%) in azoles, according to CLSI criteria. Quality control was performed in parallel using *Candida parapsilosis* ATCC 22019 and *Candida krusei* ATCC 6258 to ensure accuracy and reproducibility of the assay.

MIC values were determined visually, and results were expressed as ranges and geometric means (µg/mL) for each antifungal agent. All tests were performed in triplicate.

### 2.4. Statistical Analysis

Statistical analysis was performed using GraphPad Prism v.9.0 (GraphPad Software Inc., San Diego, CA, USA). Data distribution was assessed using the Shapiro–Wilk normality test. DNA concentration and purity data were analyzed using ANOVA followed by Tukey’s multiple comparisons test. The significance level was set at *p* < 0.05.

## 3. Results

### 3.1. Clinical and Epidemiological Characteristics

The first objective of this study was to evaluate the clinical and epidemiological characteristics of domestic cats diagnosed with sporotrichosis in Manaus, Brazilian Amazon. The main findings are summarized in Table 1 and illustrated in Figure 1.

The majority of affected cats were young, with 25 animals (86.2%) aged between one and three years. Male cats predominated, representing 82.7% (24/29) of cases. In terms of behavioral classification, 44.8% (13/29) of the cats were semi-indoor and another 44.8% were free-roaming. Regarding lesion distribution, 55.2% (16/29) of the cats presented localized lesions restricted to a single anatomical site, such as the nasal planum, face, limbs, or tail. The remaining 44.8% (13/29) exhibited disseminated lesions affecting multiple anatomical regions, including combinations of the head, limbs, trunk, ears, and paw pads. The nasal planum (23.3%, 10/43), face (18.6%, 8/43), and paws (16.3%, 7/43) were the most commonly affected sites, as shown in Table 1. In summary, most affected cats were young males with free-roaming behavior, and lesions were predominantly localized at the nasal planum, face, and limbs.

Ulcerative and nodular lesions with crusts and secretions ranging from serous to bloody were characteristic. An example of a typical clinical lesion, along with the corresponding fungal culture and microscopic morphology of *S. brasiliensis* isolated from a feline case, is presented in Figure 1.

### 3.2. DNA Extraction Testing and PCR Amplification Efficiency

The second objective of this study was to evaluate the efficiency of three DNA extraction methods applied to fungal isolates of medical importance. For this assay, we selected one regional isolate (S02) along with four well-characterized ATCC reference strains. The main results are presented in Figure 2.

Regarding DNA concentration (Figure 2A), the phenol–chloroform method yielded the highest mean DNA concentrations across all isolates, with values ranging from 30.5 ± 2.1 ng/µL for *Sporothrix schenckii* ATCC CFP-00746 to 47.8 ± 1.7 ng/µL for *Aspergillus niger* ATCC 16404. The Silica Column method produced intermediate DNA concentrations, with values between 12.4 ± 1.3 ng/µL and 28.9 ± 1.6 ng/µL. The Acetate Precipitation method showed the lowest DNA yields, with concentrations ranging from 4.2 ± 0.8 ng/µL to 9.5 ± 1.2 ng/µL. In terms of DNA purity (Figure 2B), the phenol–chloroform and Silica Column methods provided the highest 260/280 absorbance ratios, ranging from 1.6 ± 0.1 to 1.9 ± 0.1 across the different isolates, while the Acetate Precipitation method showed lower purity values, ranging between 1.3 ± 0.1 and 1.5 ± 0.1.

To complement the DNA extraction analysis, the amplification efficiency of three primer pairs targeting different regions of the ribosomal DNA of *S. brasiliensis* S02 was evaluated. The *ITS1*–*ITS4* primer pair demonstrated a successful amplification of the target region at all tested DNA concentrations, ranging from 20 ng to 0.0002 ng. The *NL1*–*NL4* primers also showed good amplification efficiency, with visible and well-defined bands down to 0.002 ng. The *ITS5*–*NL4* primers produced amplification products with satisfactory band intensity at DNA concentrations as low as 0.02 ng, indicating lower sensitivity compared to the other primer sets.

Overall, the phenol–chloroform method produced significantly higher DNA concentrations and purity compared to other methods (*p* < 0.05), and *ITS1*–*ITS4* and *NL1*–*NL4* primers demonstrated the best sensitivity for PCR detection within the conducted comparison.

### 3.3. In Silico RFLP Analysis

To evaluate the discriminatory potential of different restriction enzymes applied to the *ITS* region of fungal species, an in silico analysis was performed using the NEBcutter V3.0 software with TYPE strain sequences deposited in GenBank. The predicted fragment sizes generated by the enzymes *HaeIII*, *MspI*, *HinfI*, *DdeI*, and *RsaI* for *Sporothrix brasiliensis CBS* 120339, *Sporothrix schenckii* CBS 359.36, *Sporothrix globosa* CBS 120340, *Aspergillus niger* ATCC 16888, and *Candida albicans* CBS 562 are presented in Table 2.

Among the in silico-assessed *Sporothrix* species, the digestion profiles obtained with each enzyme had highly similar bp. For example, digestion with *HaeIII* produced five main fragments for *S. brasiliensis* (82, 33, 276, 21, and 116 bp), *S. schenckii* (81, 33, 276, 21, and 114 bp), and *S. globosa* (97, 33, 277, 21, and 113 bp). *MspI* generated fragment pairs of 463–65 bp, 462–63 bp, and 479–62 bp for these same species, respectively. Similarly, *HinfI* produced profiles consisting of five fragments, with the largest fragments ranging between 253 and 258 bp. Although these differences were detected in silico, most fragment sizes were very close, with variations under 20 bp, which may limit the practical differentiation in electrophoresis of these species applying PCR-RFLP. In contrast, *A. niger* and *C. albicans* exhibited distinct fragmentation patterns for all tested enzymes, as observed with *HaeIII* (six fragments for *A. niger*, two fragments of 69 and 426 bp for *C. albicans*) and *DdeI* (187–389 bp for *A. niger* and 397–98 bp for *C. albicans*), allowing their clear differentiation from the *Sporothrix* spp. group.

In summary, in silico RFLP profiles showed minimal differences among *Sporothrix* species, demonstrating the limited discriminatory power of this method under practical conditions.

### 3.4. Phylogenetic Analysis

The fourth objective of this study was to confirm the phylogenetic identity of four feline Sporothrix isolates from Manaus using internal transcribed spacer (*ITS*)-based analysis. Although more isolates were initially processed (*n* = 29), in our described experimental conditions, only four produced *ITS* amplicons of sufficient quality for sequencing. The *ITS* region was selected based on the broad acceptance of *ITS* for species-level identification of *Sporothrix* spp. [8]. The neighbor-joining tree was constructed in MEGA X (version 10) with 1000 bootstrap replicates and is shown in Figure 3. All four isolates are grouped within the *S. brasiliensis* isolates.

The analysis included *ITS* sequences of reference strains of *S. brasiliensis*, *S. schenckii*, and *S. globosa* obtained from the NCBI database. All four regional isolates from the Amazon are clustered within the *S. brasiliensis* cluster, along with the reference strains. The isolates used as outgroups, *Aspergillus niger* ATCC 16888 and *Candida albicans* CBS 562, are grouped separately and are highlighted in red in Figure 3. The regional isolates analyzed in this study are shown in green. All sequences used in this analysis are publicly available in the NCBI database (https://www.ncbi.nlm.nih.gov/genbank/): Sample 36-PV991410; Sample 03-PV991411; Sample 34-PV991412; Sample 02-PV991379; accessed on 6 July 2025.

### 3.5. Antifungal Susceptibility Profile (MIC Determination)

To evaluate in vitro drug susceptibility, minimum inhibitory concentrations (MICs) were determined for 29 *S. brasiliensis* isolates using the CLSI broth microdilution method. The assays included amphotericin B, itraconazole, ketoconazole, and fluconazole. Results are summarized in Table 3. In short, antifungal susceptibility testing indicated low MIC values for itraconazole and ketoconazole, and consistently high MICs for fluconazole. All isolates exhibited MIC values of 64 µg/mL for fluconazole, without variation observed among tested samples.

## 4. Discussion

This study presents clinical, epidemiological, and molecular data on feline sporotrichosis cases in the Amazon region, contributing to diagnostic standardization. Clinically, most affected cats were young males exhibiting ulcerative and nodular lesions, frequently on the nasal planum, face, and limbs, with either localized or disseminated distribution. The phenol–chloroform extraction method yielded the highest DNA quality and concentration across isolates. Minimum DNA concentrations for successful amplification were defined using pan-fungal primer pairs *ITS1*–*ITS4*, *NL1*–*NL4*, and *ITS5*–*NL4*. Although *ITS*-based PCR-RFLP showed limitations in species discrimination, *ITS* sequencing enabled the identification of *Sporothrix brasiliensis* in feline isolates. This represents one of the first regional applications of *ITS* sequencing in the Amazon. Preliminary antifungal susceptibility data indicated low MIC values for itraconazole.

The predominance of young (1–3 years; 86.2%) male (82.7%) cats exhibiting territorial or free-roaming behavior observed in this study aligns with the epidemiological profile described in other endemic regions. Gremião et al. [5], studying feline sporotrichosis in Rio de Janeiro, reported that most affected animals were young, outdoor-dwelling, and aggressive males, frequently engaged in fighting, which facilitates transmission through scratches and bites. Similarly, Etchecopaz et al. [20], analyzing emerging cases in Argentina, highlighted the role of male territorial behavior in the dissemination of *S. brasiliensis*. Rodrigues et al. [2] further emphasized that lesions commonly affect the head, face, limbs, and tail—areas associated with combat wounds—supporting the behavioral hypothesis of transmission. In our cohort, the frequent involvement of the nasal planum, forelimbs, and ears reinforces this pattern, suggesting that behavioral traits and anatomical exposure contribute significantly to disease progression and lesion distribution.

Regarding DNA extraction, our findings demonstrated superior efficiency of the phenol–chloroform method, yielding significantly higher DNA concentrations and purity compared to Silica Column and Acetate Precipitation methods. These results are corroborated by previous studies [16,21], which also highlighted the effectiveness of phenol–chloroform in removing proteins and contaminants. The consistent purity values (260/280 ratio between 1.6 and 2.0) obtained confirmed the method’s suitability for nucleic acid amplification-based assays. Nonetheless, the toxic nature of phenol–chloroform necessitates careful handling, suggesting safer alternatives, such as silica columns. Despite their slightly lower DNA yield, such columns could be more practical in clinical diagnostic routine.

In silico PCR-RFLP analyses showed that commonly used restriction enzymes like *HaeIII*, *MspI*, *HinfI*, and *DdeI* produced highly similar fragment patterns among *Sporothrix* species, with limited fragment size differences (less than 20 bp). This finding aligns with earlier studies [4,22,23], emphasizing PCR-RFLP’s limitations due to minor fragment size variations, making this approach insufficient for reliable species differentiation in practical scenarios. Genes such as β-tubulin and calmodulin have demonstrated greater accuracy in differentiating *Sporothrix* species [4,22,23] and could be integrated in research workflows.

Phylogenetic analysis based on the internal transcribed spacer (*ITS*) region was used to confirm the molecular identity of *Sporothrix* isolates recovered from feline cases in the Amazon region. Although multi-locus sequencing schemes have been proposed to improve resolution within the *Sporothrix* complex, several studies continue to support the discriminatory potential of the *ITS* region. Rodrigues et al. [24] and Zhou et al. [8] highlighted its ability to separate the main clinically relevant species, such as *S. brasiliensis*, *S. schenckii*, and *S. globosa*, using *ITS*-based phylogenies. More recently, Carvalho et al. [25] reinforced its practical application by demonstrating *ITS* clustering consistency with species assignment. In this study, all four feline isolates analyzed showed high-quality *ITS* amplification and sequencing, clustering consistently within the *S. brasiliensis* clade. The limited number of sequences obtained (4/29 isolates) is primarily attributable to difficulties in *ITS* amplification in approximately 70% of the samples and to the suboptimal quality of extracted DNA in others. These findings are in line with previous reports of the predominance of *S. brasiliensis* as the primary etiological agent in feline sporotrichosis across Brazil [4,25,26,27]. The widespread use of *ITS* in diagnostic laboratories, coupled with *ITS* representation in public databases, supports the continued relevance of *ITS* as a phylogenetic marker for initial species delimitation in *Sporothrix* infections.

Antifungal susceptibility testing of *S. brasiliensis* isolates showed MIC values consistent with previous studies. The geometric mean MIC for itraconazole (0.57 µg/mL) and ketoconazole (0.25 µg/mL) were comparable to those previously reported [6,28]. Amphotericin B assessment even showed higher MICs (7.27 µg/mL) compared to the results published by Etchecopaz et al. [20]. Fluconazole consistently exhibited limited activity (64 µg/mL), aligning with previously documented resistance profiles [2,6]. These findings highlight regional consistency of low itraconazole MIC values.

Sporotrichosis is an emerging fungal infection which has been a cause of growing concern in Brazil since the late 1990s. Now, it represents the most prevalent and most widely distributed implantation mycosis globally. In Manaus, Amazonas State, surveillance data recorded 4301 confirmed cases between August 2020 and December 2023, including 3403 animal cases (99.6% in cats) and 898 human cases, with cases reported across all districts of the city. During this period, human cases doubled and animal cases quadrupled, with a clear spatial association between animal and human cases [29]. These findings highlight an urgent need for coordinated veterinary and municipal monitoring programs to enable early detection, improve epidemiological mapping, and support targeted interventions to reduce transmission to humans. Although transmission of *S. brasiliensis* in Manaus is primarily linked to direct contact with infected cats, the possibility of indirect or nosocomial transmission should also be further investigated. Recent reports highlight environmental contamination in veterinary clinics and fomites as potential sources of infection, suggesting a need for strict biosafety measures [6].

This study presents several limitations that should be acknowledged. First, the relatively small number of analyzed isolates restricts the generalizability of the findings and may not fully represent the genetic variability of *Sporothrix* spp. in the Amazon region. Second, although the phenol–chloroform method proved effective for DNA extraction, its known toxicity limits its applicability in routine clinical settings, highlighting a need for safer alternatives. Third, DNA quality assessment was based solely on yield and purity. Further analyses focusing on DNA integrity and amplification efficiency would strengthen the conclusions. Additionally, only the *ITS* region was used for molecular identification, which, despite its broad use and database representation, has demonstrated limitations in discriminating closely related species within the *Sporothrix* complex. The absence of multi-locus approaches using markers such as β-tubulin or calmodulin may have constrained the phylogenetic resolution.

## 5. Conclusions

This study contributed to the assessment of the ongoing feline sporotrichosis outbreak in the Brazilian Amazon and confirmed *S. brasiliensis* as the predominant species involved. In particular, it provided details on clinical and epidemiological features of the affected cats, supporting available case definitions. The study’s contributions to the standardization of molecular diagnostic approaches may support regional species identification. In detail, the phenol–chloroform method demonstrated the best DNA extraction efficiency in the conducted comparison, and the *ITS1*–*ITS4* primers were adequate for PCR detection. Although PCR-RFLP showed limited discriminatory power within the *Sporothrix* genus, phylogenetic analysis based on *ITS* sequences allowed species identification by clustering within the *S. brasiliensis* clade. These findings may contribute to the standardization of the molecular diagnosis of *S. brasiliensis* as well as to the epidemiological surveillance of sporotrichosis in endemic regions, supporting optimized disease prevention and control strategies.

## Figures and Tables

**Figure 1 animals-15-02318-f001:**
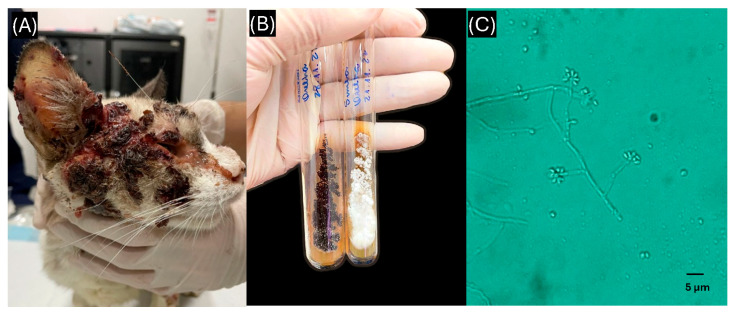
Clinical lesion, fungal culture, and microscopic morphology of *Sporothrix brasiliensis* isolated from a feline case. (**A**) Feline with a lesion on the lateral facial region showing cutaneous involvement with dissemination to the ocular area, ear, and nasal planum swelling; (**B**) colony of *S. brasiliensis* grown on Sabouraud agar after incubation, obtained from a sample collected from the ocular and ear regions; (**C**) microscopic morphology of *S. brasiliensis* displaying the characteristic “daisy-like” arrangement, observed at 400× magnification.

**Figure 2 animals-15-02318-f002:**
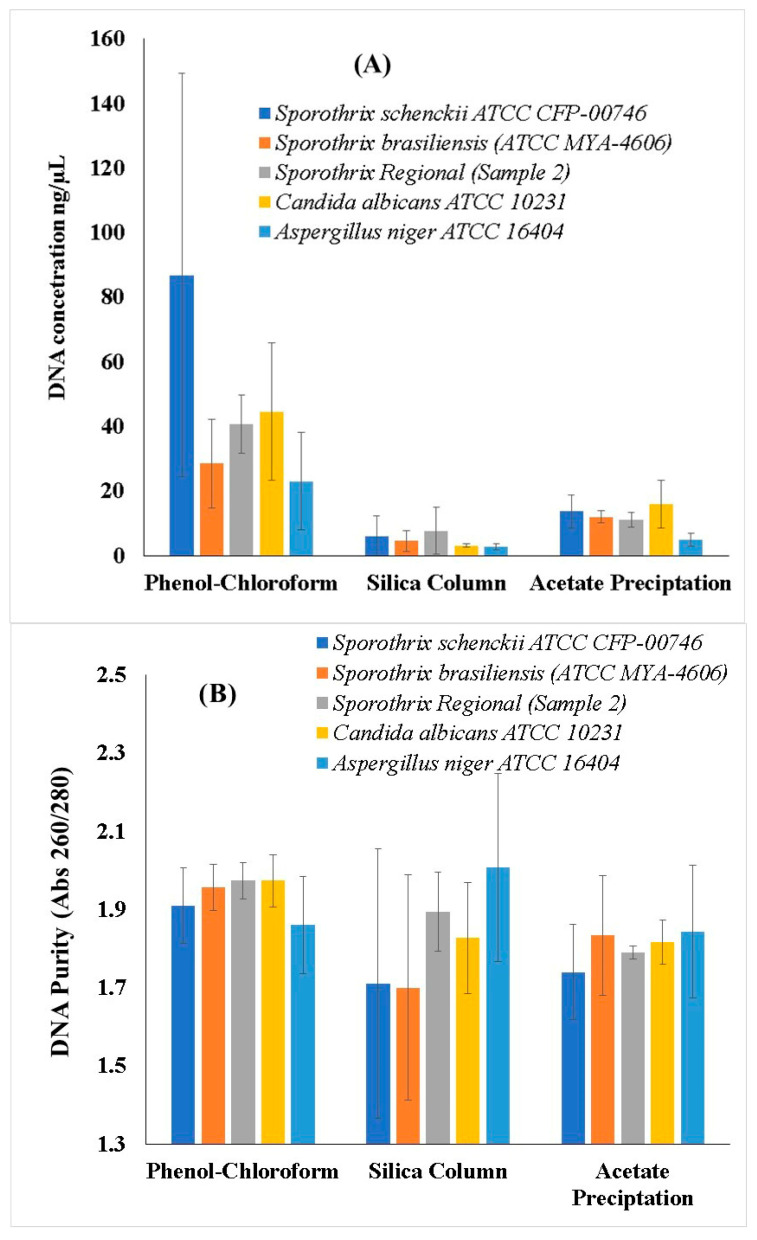
Comparative evaluation of DNA extraction methods applied to fungal isolates of *Sporothrix schenckii* ATCC CFP-00746, *Sporothrix brasiliensis* ATCC MYA-4606, a regional Sporothrix isolate (Sample 2), *Candida albicans* ATCC 10231, and *Aspergillus niger* ATCC 16404. (**A**) Mean DNA concentration (ng/µL) obtained using the phenol–chloroform, Silica Column, and Acetate Precipitation methods. (**B**) Mean DNA purity, expressed as the 260/280 nm absorbance ratio. Error bars represent the standard deviation from measurements performed in triplicate.

**Figure 3 animals-15-02318-f003:**
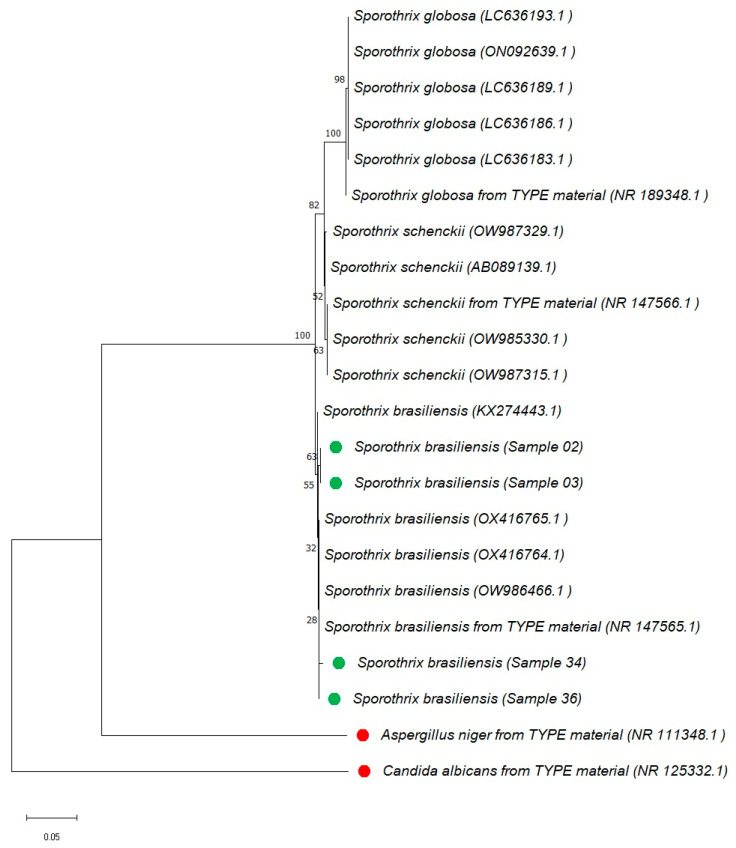
Phylogenetic analysis based on the ITS region using the neighbor-joining method with bootstrap support, performed applying the MEGA (Molecular Evolutionary Genetics Analysis) X software package. The tree was constructed using the internal transcribed spacer (*ITS*) molecular marker, based on the alignment of *Sporothrix* species sequences available in the NCBI database. Regional isolates from the Amazon region included in this study are shown in green. *Aspergillus niger* and *Candida albicans* were used as outgroup controls and are shown in red. All sequence data are available in the NCBI database.

**Table 1 animals-15-02318-t001:** Epidemiological and clinical characteristics of domestic cats diagnosed with sporotrichosis in a veterinary clinic of spontaneous demand, Manaus, Brazilian Amazon.

Epidemiological and Clinical Characteristics	Sporotrichosis Positive Isolates (%)
**Age**	
6 months	1/29 (3.4%)
8 months	1/29 (3.4%)
1 year	6/29 (20.7%)
2 years	14/29 (48.3%)
3 years	5/29 (17.2%)
4 years	1/29 (3.4%)
7 years	1/29 (3.4%)
**Sex**	
Male	24/29 (82.7%)
Female	5/29 (17.2%)
**Behavioral features**	
Indoor	3/29 (10.3%)
Semi-indoor	13/29 (44.8%)
Free-roaming	13/29 (44.8%)
**Clinical Manifestation**	
Localized lesions (lesions confined to a single anatomical region)	16/29 (55.2%)
Disseminated lesions	13/29 (44.8%)
**Anatomical distribution of all reported lesions ^1^**	
Nasal planum	10/43 (23.3%)
Face (excluding nasal planum, eyes and ears)	8/43 (18.6%)
Paws (including forelimbs and hindlimbs)	7/43 (16.3%)
Ribs and thoracic flank	4/43 (9.3%)
Ears (auricle)	3/43 (7.0%)
Tail	3/43 (7.0%)
Eye and periocular region	1/43 (2.3%)
Penile region	1/43 (2.3%)
Post-surgical abdominal site	1/43 (2.3%)
Dorsum (thoracolumbar)	1/43 (2.3%)
Paw pads	2/43 (4.7%)
Total number of lesion sites identified	43/29 cats ^2^

^1^ Some cats presented lesions in more than one anatomical region. ^2^ The total number of lesion sites exceeds the number of animals due to multifocal skin involvement in 13/29 cats.

**Table 2 animals-15-02318-t002:** In silico predicted fragment sizes (bp) from *ITS* region digestion with commercial restriction enzymes (*HaeIII*, *MspI*, *HinfI*, and *DdeI*) for five medically important fungal species: *Sporothrix brasiliensis* CBS 120339, *Sporothrix schenckii* CBS 359.36, *Sporothrix globosa* CBS 120340, *Aspergillus niger* ATCC 16888, and *Candida albicans* CBS 562. Analyses were performed using the NEBcutter V3.0 tool based on amplified *ITS* sequences, with fragment sizes expressed in base pairs (bp), aiming at species differentiation by PCR-RFLP.

Enzyme	*S. brasiliensis* CBS 120339	*S. schenckii* CBS 359.36	*S. globosa* CBS 120340	*A. niger* ATCC 16888	*C. albicans* CBS 562
*HaeIII* (GG/CC)	82, 33, 276, 21, 116	81, 33, 276, 21, 114	97, 33, 277, 21, 113	47, 45, 15, 29, 291, 73, 76	69, 426
*MspI* (C/CGG)	463, 65	462, 63	479, 62	111, 21, 15, 93, 101, 42, 32, 86, 75	276, 219
*HinfI* (G/ANTC)	130, 60, 16, 64, 258	129, 61, 16, 64, 255	145, 63, 16, 64, 253	19, 208, 16, 56, 248, 29	162, 16, 56, 8, 137, 116
*DdeI* (C/TNAG)	154, 374	153, 372	169, 372	187, 389	397, 98
*BsaI* (G/TAC)	22, 415, 91	22, 414, 89	41, 412, 88	73, 503	495

**Table 3 animals-15-02318-t003:** Minimum inhibitory concentrations (MICs) of antifungal agents against *Sporothrix brasiliensis* isolates obtained from feline clinical cases in veterinary clinics in Manaus (Brazilian Amazon).

Antifungal Agent	MIC Range (µg/mL)	Geometric Mean (µg/mL)
Ketoconazole	0.125–0.5	0.25
Itraconazole	0.25–1	0.57
Amphotericin B	4–8	7.27
Fluconazole	64	64

## Data Availability

The data presented in this study are available on request from the corresponding author. The *ITS* sequences of *S. brasiliensis* obtained from feline clinical isolates have been deposited in the NCBI GenBank database (https://www.ncbi.nlm.nih.gov/genbank/) under the following accession numbers: Sample 36-PV991410; Sample 03-PV991411; Sample 34-PV991412; Sample 02-PV991379, accessed on 6 July 2025.

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
