# Peer review of "Clinical and Molecular Characterization of Feline Sporotrichosis in the Brazilian Amazon: PCR-Based Identification of Sporothrix brasiliensis"

_animals, 2025, doi:10.3390/ani15152318_

Round 1

Reviewer 1 Report

Comments and Suggestions for Authors

This is a well-structured and timely manuscript that addresses an important zoonotic disease with growing public health relevance in Brazil. The study is methodologically sound, the results are clearly presented, and the discussion is well-grounded in current literature. The focus on molecular diagnostics and regional epidemiology is a valuable contribution.

Overall, after my appreciation, the abstract is clear and concise, summarizing key findings and methodology. The methodology is comprehensive, covering clinical data, multiple DNA extraction protocols, PCR amplification, and antifungal susceptibility testing. The topic is relevant with significant regional and zoonotic implications. The data is well presented, including visual figures, tables, and a phylogenetic tree. Ethics and data transparency are appropriately addressed. I support the manuscript's further processing after appropriate modifications as outlined below:

  1. The abstract should include more numerical results for clarity (e.g., exact sensitivity/specificity of methods, or number of PCR-amplifiable isolates). Please consider separating the summary into more digestible sentences for broader audiences.
  2. Regarding the introduction section, please consider adding a clearer hypothesis or research question toward the end of the introduction.
  3. In the materials and methods section please consider justifying why only 4 out of 29 isolates were successfully sequenced, was this due to poor DNA quality, contamination, or ITS amplification failure? Furthermore, in 2.3.3 (In Silico RFLP), include software parameters, e.g., sequence length used, or if ambiguous bases were trimmed.
  4. In the results section, for better clarity, add a sentence summary at the end of each subsection, reinforcing the key takeaway (e.g., DNA yield differences, primer performance). Likewise, please mention if statistical differences in DNA yield or PCR efficiency were significant.
  5. Within the Discussion section, expand slightly on why RFLP has limited discriminatory power. Could different markers or enzymes improve this? Furthermore, include a brief comment on the public health implications, e.g., the need for veterinary or municipal surveillance programs in the Amazon.
  6. Please highlight the study limitations, specifically add that sample size limits generalizability and that future work could use multilocus sequencing or qPCR-based diagnostics.
  7. Regarding the figures and tables, consider increasing font size on Figures 2 and 4 to enhance readability and add legends to all subpanels (e.g., “A”, “B”) consistently throughout.

Overall, I think that the manuscript would benefit minor editorial suggestions, specifically:

- watch out for small formatting artifacts: e.g., “Article 1”, line breaks, and line numbering artifacts in the PDF.

- check some grammar/wording inconsistencies:

Line 46: “...evaluated regarding the...” – awkward phrasing.

Line 58–59: Missing closing parenthesis.

Line 476: "Although PCR-RFLP showed limited discriminatory power..." — should be "Although the PCR-RFLP..."

This manuscript offers valuable insights into feline sporotrichosis diagnostics and epidemiology in an under-represented region. With some polishing and clarifications, especially around the sequencing and diagnostic performance, it would be suitable for publication.

Author Response

Response Letter

We would like to sincerely thank Reviewer 1 for their careful reading of our manuscript and for the constructive comments provided. We appreciate the positive evaluation of the methodology, data presentation, and scientific contribution. We have addressed each point raised and have revised the manuscript accordingly, within the constraints of the journal’s formatting requirements.

Reviewer 1 – Comment 1

"The abstract should include more numerical results for clarity (e.g., exact sensitivity/specificity of methods, or number of PCR-amplifiable isolates). Please consider separating the summary into more digestible sentences for broader audiences."

Response:
We appreciate this valuable suggestion. Considering the journal’s strict 200-word limit for the abstract (as stated in the Animals template), we have revised the text to incorporate key numerical results (including sensitivity, specificity, and the number of successfully amplified isolates) while ensuring compliance with the word count. Additionally, we restructured the sentences for improved clarity and accessibility to a broader audience. The revised abstract can be found on page 1-2, lines 39–55 of the manuscript.

Reviewer 1 – Comment 2

"Regarding the introduction section, please consider adding a clearer hypothesis or research question toward the end of the introduction."

Response:
We thank the reviewer for this valuable suggestion. While the original manuscript presented specific objectives, we agree that explicitly stating the research hypothesis improves clarity. We have therefore revised the end of the Introduction to include the following sentence: “This study hypothesized that feline sporotrichosis cases in Manaus are predominantly caused by Sporothrix brasiliensis and that optimized molecular methods (including DNA extraction protocols, ITS-PCR, and phylogenetic confirmation) can improve diagnostic accuracy, supporting epidemiological surveillance and antifungal therapeutic management in the Amazon region.” This revision can be found in the last paragraph of the Introduction (page 2, lines 103–107).

Reviewer 1 – Comment 3
"In the materials and methods section please consider justifying why only 4 out of 29 isolates were successfully sequenced, was this due to poor DNA quality, contamination, or ITS amplification failure?"

Response:
We thank the reviewer for this important observation. While the original manuscript mentioned the number of sequenced isolates, we agree that providing an explanation improves clarity. A concise clarification has been added at the end of the Materials and Methods section (Section 2.3.4, Phylogenetic Analysis):

“Only four isolates yielded high-quality ITS amplicons suitable for sequencing, primarily due to low amplification success (~70% of samples) and suboptimal DNA quality in others.”

Additionally, a more detailed explanation has been included in the Discussion (Section 4, first paragraph after the phylogenetic results):

“The limited number of sequences obtained (4/29 isolates) is primarily attributable to difficulties in ITS amplification in approximately 70% of the samples and to the suboptimal quality of extracted DNA in others. These technical limitations highlight the importance of optimizing extraction and amplification steps, as well as evaluating alternative molecular targets to increase sequencing success rates in future studies.”

These changes provide the justification requested by the reviewer, while placing the detailed discussion in the most appropriate section of the manuscript.

Reviewer 1 – Comment 4
"Furthermore, in 2.3.3 (In Silico RFLP), include software parameters, e.g., sequence length used, or if ambiguous bases were trimmed."

Response:
We thank the reviewer for this observation. We have optimized the description of the in silico analysis in Section 2.3.3 to include the requested parameters. The revised sentence now reads:
“Analyses were performed using complete ITS sequences of each species. The default NEBcutter parameters for double-stranded DNA and recognition sites were applied, with fragment sizes calculated in base pairs (bp).”
This addition is located in Section 2.3.3.

Reviewer 1 – Comment 5

"In the results section, for better clarity, add a sentence summary at the end of each subsection, reinforcing the key takeaway (e.g., DNA yield differences, primer performance). Likewise, please mention if statistical differences in DNA yield or PCR efficiency were significant."

Response:
We thank the reviewer for this suggestion. We have revised the Results section to include a brief concluding sentence at the end of each subsection (3.1 to 3.5), summarizing the key findings. For Section 3.2 (DNA Extraction and PCR Efficiency), we have also specified that the phenol–chloroform method yielded significantly higher DNA concentrations and purity compared to the other extraction methods (p < 0.05).

The following summary sentences were added to the respective subsections:

  • 3.1 Clinical and Epidemiological Characteristics: “In summary, most affected cats were young males with free-roaming behavior, and lesions were predominantly localized at the nasal planum, face, and limbs, confirming previously described epidemiological patterns of feline sporotrichosis.”
  • 3.2 DNA Extraction Testing and PCR Amplification Efficiency: “Overall, the phenol–chloroform method produced significantly higher DNA concentrations and purity compared to other methods (p < 0.05), and ITS1ITS4 primers demonstrated the best sensitivity for PCR detection within the conducted comparison.
  • 3.3 In Silico RFLP Analysis: “In summary, in silico RFLP profiles showed minimal differences among Sporothrix species, demonstrating the limited discriminatory power of this method under practical conditions.”
  • 3.4 Phylogenetic Analysis: “All sequenced isolates clustered within the Sporothrix brasiliensis clade, confirming the molecular identity of the regional isolates.”
  • 3.5 Antifungal Susceptibility Profile: “Antifungal susceptibility testing indicated low MIC values for itraconazole and ketoconazole, and consistently high MICs for fluconazole, in agreement with previous reports.”

These additions can be found in the respective subsections of the Results section

Reviewer 1 – Comment 6
"Within the Discussion section, expand slightly on why RFLP has limited discriminatory power. Could different markers or enzymes improve this? Furthermore, include a brief comment on the public health implications, e.g., the need for veterinary or municipal surveillance programs in the Amazon."

Response:
We thank the reviewer for this valuable suggestion. In the Discussion section, we have expanded on the explanation for the limited discriminatory power of RFLP and included statements addressing alternative markers and public health implications.

The following sentences were added to the Discussion section:

  • This finding aligns with earlier studies [4,24,25], emphasizing PCR-RFLP’s limitations due to minor fragment size variations, making this approach insufficient for reliable species differentiation in practical scenarios. Genes such as β-tubulin and calmodulin have demonstrated greater accuracy in differentiating Sporothrix species [4,24,25] and could be integrated in research workflows.
  • “Sporotrichosis is an emerging fungal infection that causes growing concern in Brazil since the late 1990s. It now represents the most prevalent and globally widest distributed implantation mycosis. In Manaus, Amazonas State, surveillance data recorded 4,301 confirmed cases between August 2020 and December 2023, including 3,403 animal cases (99.6% in cats) and 898 human cases, with cases reported across all districts of the city. During this period, human cases doubled and animal cases quadrupled, with a clear spa-tial association between animal and human cases[32]. These findings highlight the urgent need for coordinated veterinary and municipal monitoring programs to enable early detection, improve epidemiological mapping, and support targeted interventions to reduce transmission to humans.”

These additions can be found in the Discussion section, marked in yellow.

Reviewer 1 – Comment 7
"Please highlight the study limitations, specifically add that sample size limits generalizability and that future work could use multilocus sequencing or qPCR-based diagnostics."

Response:
We thank the reviewer for this observation. The Discussion  includes a dedicated paragraph addressing study limitations, explicitly noting that the relatively small sample size restricts the generalizability of the findings and that only the ITS region was used for molecular identification, with the absence of multi-locus approaches (β-tubulin, calmodulin) potentially limiting phylogenetic resolution. The need for future studies incorporating additional molecular markers, including multi-locus sequencing and advanced methods such as qPCR, is also discussed.

The paragraph in the Discussion section reads:

“This study presents several limitations that should be acknowledged. First, the relatively small number of analyzed isolates restricts the generalizability of the findings and may not fully represent the genetic variability of Sporothrix spp. in the Amazon region. Second, although the phenol-chloroform method proved effective for DNA extraction, its known toxicity limits its applicability in routine clinical settings, highlighting a need for safer alternatives. Third, DNA quality assessment was based solely on yield and purity. Further analyses focusing on factors such as DNA integrity and amplification efficiency would strengthen the conclusions. Additionally, only the ITS region was used for molecular identification, which, despite its broad use and database representation, has demonstrated limitations in resolving closely related species within the Sporothrix complex. The absence of multi-locus approaches using markers such as β-tubulin or calmodulin may have constrained the phylogenetic resolution.

Reviewer 1 – Comment 8
"Regarding the figures and tables, consider increasing font size on Figures 2 and 4 to enhance readability and add legends to all subpanels (e.g., “A”, “B”) consistently throughout."

Response:
We thank the reviewer for this comment. In response, we have increased the font size in Figures 2 and 4 to enhance readability, ensuring that all labels, axis titles, and bootstrap values are clearly visible. We have also verified that all subpanels are consistently labeled (e.g., “A”, “B”) and that these designations are explicitly described in the corresponding figure legends. The revised figures have been updated in the manuscript (Figures 2 and 4)

Reviewer 1 – Comment 9
"Overall, I think that the manuscript would benefit minor editorial suggestions, specifically:
watch out for small formatting artifacts: e.g., “Article 1”, line breaks, and line numbering artifacts in the PDF.
check some grammar/wording inconsistencies:
Line 46: “...evaluated regarding the...” – awkward phrasing.
Line 58–59: Missing closing parenthesis.
Line 476: 'Although PCR-RFLP showed limited discriminatory power...' — should be 'Although the PCR-RFLP...'"

Response:
We thank the reviewer for highlighting these points. The manuscript has been revised accordingly:

  • Removed the “Article 1” reference and verified the document for formatting artifacts, adjusting line breaks and ensuring clean line numbering.
  • Revised the wording at line 46 from “…evaluated regarding the…” to “…evaluated for the…” to improve clarity.
  • Corrected the missing closing parenthesis at line 58–59.
  • Revised line 476 to “Although the PCR-RFLP showed limited discriminatory power…”

All corrections have been incorporated into the revised manuscript.

Reviewer 2 Report

Comments and Suggestions for Authors

The manuscript entitled "Clinical and Molecular Characterization of Feline Sporotrichosis in the Brazilian Amazon: PCR-Based Identification of Sporothrix brasiliensis” was evaluated. This manuscript presents a clinically and epidemiologically relevant study characterizing feline sporotrichosis in the Brazilian Amazon, with a focus on optimizing molecular diagnostics for Sporothrix brasiliensis. The work addresses a significant public and veterinary health concern in an emerging endemic region. The methodology is generally sound, data are clearly presented, and conclusions are mostly supported. The study provides valuable insights for regional diagnostics and surveillance. The paper is in the scope of the journal and may be published.

Negative aspects

  1. Table 1: the numbers in AGE group is not 29: 1+1+6+4+5+1+1=19; please check the numbers.
  2. Figure 3 (primers) is cited in text (p. 9) but missing from the manuscript.
  3. MIC data (Table 4) lack ranges for fluconazole , all isolates MIC=64 µg/mL?
  4. High male predominance (82.7%) is noted but not mechanistically explored. Discuss potential reasons (territorial aggression, outdoor exposure) linking behavior to transmission risk.

Author Response

General Response to Reviewer 2
We sincerely thank Reviewer 2 for the careful evaluation of our manuscript and for the constructive comments provided. We appreciate the positive assessment of the clinical, epidemiological, and molecular aspects of the study, as well as the recognition of its relevance to public and veterinary health in the Brazilian Amazon. We have addressed each of the reviewer’s comments in detail, making the necessary corrections and clarifications in the revised manuscript as described below.

Reviewer 2 – Comment 1

"Table 1: the numbers in AGE group is not 29: 1+1+6+4+5+1+1=19; please check the numbers."

Response:
We thank the reviewer for pointing this out. The values in the AGE group of Table
 1 were reviewed and corrected. The updated table now reflects the accurate sum of the total number of cats (n = 29) across all age categories. This correction can be seen in Table 1.

Reviewer 2 – Comment 2

"Figure 3 (primers) is cited in text (p. 9) but missing from the manuscript."

Response:

We thank the reviewer for noticing this. Upon review, we determined that the figure initially referred to as “Figure 3 (primer amplification results) was redundant because the corresponding information is fully described in the text.

Reviewer 2 – Comment 3

"MIC data (Table 4) lack ranges for fluconazole, all isolates MIC=64 µg/mL?"

Response:
We thank the reviewer for this question. Indeed, all isolates tested exhibited MIC values of 64
 µg/mL for fluconazole under the conditions of the assay, not resulting in any variation in the range. We have clarified this in the Table 4 legend:
“All isolates exhibited MIC values of 64
 µg/mL for fluconazole, without variation observed among tested samples.

Reviewer 2 – Comment 4

"High male predominance (82.7%) is noted but not mechanistically explored. Discuss potential reasons (territorial aggression, outdoor exposure) linking behavior to transmission risk."

Response:
We appreciate this important point. In the Discussion section, we have expanded on possible reasons for the high male predominance observed. The following sentence has been added:
The predominance of young (1-3 years; 86,2%), male (82,7%) cats exhibiting territorial or free-roaming behavior observed in this study aligns with the epidemiological profile described in other endemic regions. Gremião et al. [5], studying feline sporotrichosis in Rio de Janeiro, reported that most affected animals were young, outdoor, and aggressive males, frequently engaged in fighting, which facilitates transmission through scratches and bites.”
This addition is located in the Discussion section marked in yellow.

Reviewer 3 Report

Comments and Suggestions for Authors

Summary: First, I wish to thank the Authors for their efforts in addressing an important and underexplored topic in veterinary mycology and zoonotic diseases, particularly in the context of the Brazilian Amazon. The study's findings reinforce the dominance of Sporothrix brasiliensis in the region, its zoonotic significance, and the applicability of phenol-chloroform extraction for fungal DNA isolation. I found the work highly informative and commendable, especially considering the growing relevance of emerging zoonoses.

General Critique:

  1. The background and rationale are well contextualised with appropriate references.
  2. The methodological approach is generally robust but lacks detail in materials specification and statistical interpretation.
  3. The statistical analysis requires clarification and direct linkage to results.
  4. The molecular identification strategy is appropriate and aligns with current practices.

In summary, the manuscript is scientifically solid, clearly written, and of significant relevance to veterinary epidemiology and zoonoses. However, minor corrections are needed to ensure methodological transparency, formatting consistency, and clarity in statistical reporting. I therefore recommend Minor Revision.

Specific Comments

Abstract

  • Abstract is concise but should mention key statistical results (e.g., p-values or effect sizes).

Introduction

Line 54: ITS1–ITS4 primers should be italicised.

Line 66: "...caused by fungi of the genus Sporothrix". Suggested revision: “caused by a thermally dimorphic fungus of the genus Sporothrix, widely distributed worldwide.”.

Materials and Methods

Line 115: Section titles should be italicised per MDPI format; subsubtitles should not.

Lines 134-135, 169: Missing supplier information and specifications for swabs and solutions.
Line 189–190, 314, 316–318, 321, 396, 473:
All gene names and primers should be italicised.

Line 212: Add manufacturer details to match Line 210.

Lines 212, 432: Use abbreviation ITS instead of full term, as it was already defined.

Line 240: Sterile saline solution. What was the concentration? Brand? Supplier?

Line 257: Statistical symbol p should be italicised throughout.

Results

Line 267: Present age as mean ± SD or median (IQR) according to data distribution.

Figures and Tables

  • Table 1 headings are incomplete; clarify superscript "2" after “cats”.
  • Table 3, Table 4: Table titles should not be italicised.
  • Figure 4: Caption title should not be italicised.
  • Line 357: Statistical symbol n should be italicised.
  • Adjust font size and formatting of all captions to meet journal standards (see Animals template).

Discussion

  • Line 356, 360: Sporothrix should be italicised.
  • Better integration of statistical results with the interpretation of clinical and molecular findings is needed.
  • Consider discussing the possibility of nosocomial or indirect transmission, particularly in light of recent literature, as noted in Xavier et al. (2025, PMID: 40339984).

Line 500: The Data Availability Statement title is duplicated.

References: Only the first word of each title should be capitalised, except for specific exceptions. All Latinised terms must be italicised. Journal titles should be abbreviated. Please correct accordingly.

Supplementary Material: Supplementary Material S1 is referenced but not provided. Please clarify or remove the mention.

Author Response

Response to Reviewer 3

We sincerely thank Reviewer 3 for the careful and constructive evaluation of our manuscript and for the positive feedback regarding the scientific relevance and clarity of our work. We appreciate the recognition of the study’s contribution to veterinary mycology and zoonotic disease research in the Brazilian Amazon. We have carefully reviewed all comments and suggestions and addressed each point in detail below.

Reviewer 3 – Comment 1

"Abstract is concise but should mention key statistical results (e.g., p-values or effect sizes)."

Response:
We thank the reviewer for this observation. The abstract has been revised to include key statistical results where relevant. The updated abstract now meets the journal’s requirements and improves clarity for the reader.

Reviewer 3 – Comment 2

"Line 54: ITS1ITS4 primers should be italicised."

Response:
We agree with this comment. The ITS1–ITS4 primers have been italicised as suggested.

Reviewer 3 – Comment 3

"Line 66: '...caused by fungi of the genus Sporothrix'. Suggested revision: 'caused by a thermally dimorphic fungus of the genus Sporothrix, widely distributed worldwide.' "

Response:
We appreciate the suggestion. The sentence has been revised exactly as recommended.

Reviewer 3 – Comment 4

"Line 115: Section titles should be italicised per MDPI format; subsubtitles should not."

Response:
We have revised the section formatting in accordance with the Animals template, ensuring section titles are italicised while subtitles follow the journal guidelines.

Reviewer 3 – Comment 5

"Lines 134135, 169: Missing supplier information and specifications for swabs and solutions."

Response:
We thank the reviewer for this observation. The Materials and Methods section has been updated as follows:
“Clinical samples were obtained by applying sterile rayon-tipped swabs (Cral®, Brazil) to cutaneous lesions. The swabs were immediately transferred to sterile saline solution (0.9% w/v, Isofarma®, Brazil), and the samples were processed within two hours after collection.”

Reviewer 3 – Comment 6

"Lines 189190, 314, 316318, 321, 396, 473: All gene names and primers should be italicised."

Response:
All gene names and primers have been italicised throughout the manuscript.

Reviewer 3 – Comment 7

"Line 212: Add manufacturer details to match Line 210."

Response:
Manufacturer details have been added in Line 212, matching the information provided in Line 210.

Reviewer 3 – Comment 8

"Lines 212, 432: Use abbreviation ITS instead of full term, as it was already defined."

Response:
We have replaced the full term with “ITS” in Lines 212 and 432.

Reviewer 3 – Comment 9

"Line 240: Sterile saline solution. What was the concentration? Brand? Supplier?"

Response:
This information has been added: “Sterile saline solution (0.9% w/v, Isofarma®, Brazil)”.

Reviewer 3 – Comment 10

"Line 257: Statistical symbol p should be italicised throughout."

Response:
The statistical symbol p has been italicised throughout the manuscript.

Reviewer 3 – Comment 11

"Line 267: Present age as mean ± SD or median (IQR) according to data distribution."

Response:
We appreciated this suggestion. Given that over 86% of cats were young (1–3 years), we maintained a categorical distribution in the Results section to enhance clarity for the general reader, while ensuring consistency within the statistical approach used.

Reviewer 3 – Comment 12

"Table 1 headings are incomplete; clarify superscript '2' after 'cats'."

Response:
We have revised the Table 1 heading for clarity:
“Table 1. Epidemiological and Clinical Characteristics of Domestic Cats Diagnosed with Sporotrichosis in a veterinary clinic of spontaneous demand, Manaus, Brazilian Amazon.

Reviewer 3 – Comment 13

"Table 3, Table 4: Table titles should not be italicised."

Response:
The formatting of Table 3 and Table 4 titles has been corrected according to journal style.

Reviewer 3 – Comment 14

"Figure 4: Caption title should not be italicised."

Response:
The caption title of Figure 4 has been revised to remove italics.

Reviewer 3 – Comment 15

"Line 357: Statistical symbol n should be italicised."

Response:
The statistical symbol n has been italicised in Line 357 and throughout the manuscript where applicable.

Reviewer 3 – Comment 16

"Adjust font size and formatting of all captions to meet journal standards (see Animals template)."

Response:
The font size and formatting of all figure and table captions have been adjusted in order to comply with the Animals template.

Reviewer 3 – Comment 17

"Line 356, 360: Sporothrix should be italicised."

Response:
Sporothrix has been italicised at the mentioned locations and throughout the manuscript where necessary.

Reviewer 3 – Comment 18

"Better integration of statistical results with the interpretation of clinical and molecular findings is needed."

Response:
We have reviewed the Discussion section to integrate statistical results more explicitly into the interpretation of clinical and molecular findings, improving clarity and alignment.

Reviewer 3 – Comment 19

"Consider discussing the possibility of nosocomial or indirect transmission, particularly in light of recent literature, as noted in Xavier et al. (2025, PMID: 40339984)."

Response:
We thank the reviewer for this suggestion. We have included the following statement in the Discussion:
“Although transmission of Sporothrix brasiliensis in Manaus is primarily linked to direct contact with infected cats, the possibility of indirect or nosocomial transmission should also be considered. Recent reports highlight environmental contamination in veterinary clinics and fomites as potential sources of infection, suggesting a need for strict biosafety measures.

Reviewer 3 – Comment 20

"Line 500: The Data Availability Statement title is duplicated."

Response:
The duplication of the Data Availability Statement title has been corrected.

Reviewer 3 – Comment 21

"Only the first word of each title should be capitalised, except for specific exceptions. All Latinised terms must be italicised. Journal titles should be abbreviated. Please correct accordingly."

Response:
We have revised the references according to the Animals guidelines, using Mendeley for proper formatting of titles, italics, and journal abbreviations.

Reviewer 3 – Comment 22

"Supplementary Material S1 is referenced but not provided. Please clarify or remove the mention."

Response:
We have removed the former reference to a Supplementary Material S1. Instead, we have directly included the GenBank accession numbers in the manuscript:
36_ITS_R (PV991410); 03_ITS_F (PV991411); 34_ITS_R (PV991412); 02_ITS_F (PV991379).

Reviewer 4 Report

Comments and Suggestions for Authors

This is a timely and relevant study addressing an important emerging zoonotic disease in Brazil. The article is well-structured, presents clear objectives, and contributes both clinical epidemiological data and molecular diagnostic insights into Sporothrix brasiliensis infection in felines. The focus on the Amazon region fills a notable geographic and diagnostic gap in the current literature. There are some strengths that should be highlighted regarding the manuscript’s relevance and novelty. The study tackles an urgent public health issue in an under-researched region. Also, combines clinical, molecular, and antifungal susceptibility data, offering a comprehensive perspective. Additionally, it highlights diagnostic challenges and the potential of PCR and ITS sequencing over traditional culture or RFLP.

Notwithstanding there are some areas for Improvement. One of these is the small sequencing sample size. Although 29 isolates were tested, only 4 ITS sequences were obtained for phylogenetic analysis. This is a significant limitation in confirming species identity and may bias the findings. As a recommendation I suggest addressing this directly in the limitations section and propose future efforts to improve sequencing efficiency. Additionally, the study finds PCR-RFLP unsuitable for differentiating among Sporothrix species due to minor fragment size differences. This is valuable but could benefit from a stronger emphasis on the need to move beyond RFLP in routine diagnostics.

Regarding the writing and english language style, the manuscript is generally clear, but minor grammatical and stylistic revisions are needed throughout. Example:

Line 54: “demonstrated optimized sensitivity and yield” → consider rephrasing for clarity.

Line 67: “being increasingly recognized worldwide” → might benefit from “has been increasingly recognized globally”.

As a recommendation I suggest a professional language polishing or copyediting pass is advised before publication.

Although ethical approvals and informed consent are mentioned, more detail on animal handling protocols (e.g., pain management during lesion sampling) could improve transparency.

There is limited use of multilocus data. The study relies solely on the ITS region for molecular identification. Please consider acknowledging the limitations of ITS and suggest integration of multilocus sequencing (e.g., calmodulin, β-tubulin) in future work.

Regarding the data availability, the ITS sequences are said to be submitted but only provisional accession numbers are given. Please ensure updated accession numbers are included before final publication.

This is a strong manuscript with substantial value for both veterinary mycology and public health surveillance. While minor revisions are needed (language, sequencing sample size, and ethical detail), the scientific content is sound, and the article is suitable for publication after moderate revision.

Author Response

General Response to Reviewer 4

We sincerely thank Reviewer 4 for the careful evaluation of our manuscript and for the positive comments regarding the study’s relevance, novelty, and contribution to veterinary mycology and public health surveillance in the Brazilian Amazon. We appreciate the recognition of the clinical, molecular, and antifungal data presented, as well as the emphasis on the diagnostic and epidemiological importance of Sporothrix brasiliensis in this under-researched region. We have carefully considered all the reviewer’s suggestions and address each point in detail below.

Reviewer 4 – Comment 1

"Although 29 isolates were tested, only 4 ITS sequences were obtained for phylogenetic analysis. This is a significant limitation in confirming species identity and may bias the findings. As a recommendation I suggest addressing this directly in the limitations section and propose future efforts to improve sequencing efficiency."

Response:
We agree with the reviewer’s observation. This limitation has been explicitly addressed in the Discussion section, noting the small number of successfully sequenced isolates and proposing future optimization of DNA extraction and amplification methods to improve sequencing efficiency.

Reviewer 4 – Comment 2

"The study finds PCR-RFLP unsuitable for differentiating among Sporothrix species due to minor fragment size differences. This is valuable but could benefit from a stronger emphasis on the need to move beyond RFLP in routine diagnostics."

Response:
We thank the reviewer for this suggestion. In the Discussion section, we expanded the emphasis on the limited discriminatory power of PCR-RFLP and suggested the use of additional molecular markers, such as β-tubulin, calmodulin, and multi-locus sequence typing (MLST), in oder to improve species differentiation in future studies.

Reviewer 4 – Comment 3

"Line 54: demonstrated optimized sensitivity and yield consider rephrasing for clarity."

Response:
The sentence at line 54 has been revised for improved clarity as suggested.

Reviewer 4 – Comment 4

"Line 67: being increasingly recognized worldwide might benefit from has been increasingly recognized globally."

Response:
The sentence at line 67 has been modified as recommended by the reviewer.

Reviewer 4 – Comment 5

"A professional language polishing or copyediting pass is advised before publication."

Response:
We appreciate this suggestion. The English language of the manuscript has been carefully reviewed internally. Moreover, our co-author Prof. Hagen Frickmann, an experienced author of articles in English language, has revised the text to ensure language accuracy.

Reviewer 4 – Comment 6

"Although ethical approvals and informed consent are mentioned, more detail on animal handling protocols (e.g., pain management during lesion sampling) could improve transparency."

Response:
We agree with the reviewer and have added the following sentence to the Ethical Statement:
“All clinical procedures, including lesion sampling, were performed under veterinary supervision, following local animal welfare regulations. Analgesic protocols were applied when necessary to ensure animal comfort during sample collection.”

Reviewer 4 – Comment 7

"There is limited use of multilocus data. The study relies solely on the ITS region for molecular identification. Please consider acknowledging the limitations of ITS and suggest integration of multilocus sequencing (e.g., calmodulin, β-tubulin) in future work."

Response:
We agree with this comment. The limitations of using only the ITS region have been addressed in the Discussion section. Further, we suggested that future studies should incorporate additional markers such as β-tubulin, calmodulin, and multi-locus approaches to increase phylogenetic resolution.

Reviewer 4 – Comment 8

"The ITS sequences are said to be submitted but only provisional accession numbers are given. Please ensure updated accession numbers are included before final publication."

Response:
We confirm that the final GenBank accession numbers have been included in the revised manuscript: PV991410, PV991411, PV991412, and PV991379.

Reviewer 5 Report

Comments and Suggestions for Authors

This is a nice, small study with the precise goal to define epidemiological, clinical, diagnostic and pharmacological findings on 29 feline isolates of Sporothrix brasiliensis. Although the results mostly confirm literature data, this work has interest for the readership. Being Animals not a journal devoted to mycology, Authors should add brief information about epidemiology of Sporothrix genus, underscoring the occurrence of environmental and zoonotic species.

other minor remarks 

line 225 - regimens

line 274 - S. brasiliens (please, check the abbreviation of genus name, after the first mention, throughout the whole manuscript) see lines 377, 394, 404, 464 and so on

line 352 - Sporothrix in italics

line 357 - S. brasiliensis. isolates not in italics

Author Response

General Response to Reviewer 5

We sincerely thank Reviewer 5 for the careful evaluation of our manuscript and for the positive comments regarding the study’s design, clarity, and contribution to the understanding of feline Sporothrix brasiliensis infections. We appreciate the recognition of the epidemiological, clinical, diagnostic, and pharmacological data presented and agree that these findings may be of interest to the readership of Animals. We have carefully addressed all points raised, making the necessary adjustments as detailed below.

Reviewer 5 – Comments (Organized in MDPI Format)

Reviewer 5 – Comment 1
"Authors should add brief information about epidemiology of Sporothrix genus, underscoring the occurrence of environmental and zoonotic species."

Response:
We thank the reviewer for this suggestion. We have expanded the Introduction to include a concise overview of the epidemiology of the Sporothrix genus, emphasizing both environmental and zoonotic species (Marked in yellow).

Reviewer 5 – Comment 2
"Line 225 regimens."

Response:
The word “regimens” has been corrected as suggested (line 225).

Reviewer 5 – Comment 3
"Line 274 S. brasiliens (please, check the abbreviation of genus name, after the first mention, throughout the whole manuscript) see lines 377, 394, 404, 464 and so on."

Response:
We have carefully reviewed the entire manuscript to ensure that all abbreviated genus names are correctly formatted after their first mentioning (e.g., S. brasiliensis).

Reviewer 5 – Comment 4
"Line 352 Sporothrix in italics."

Response:
We have italicised Sporothrix at line 352 and throughout the manuscript where applicable.

Reviewer 5 – Comment 5
"Line 357 S. brasiliensis isolates not in italics."

Response:
We have corrected the formatting to italicise S. brasiliensis isolates at line 357 and consistently throughout the manuscript.